# Tailpipe VOC Emissions from Late Model Gasoline Passenger Vehicles in the Japanese Market

**Hiroo Hata *, Megumi Okada, Chikage Funakubo and Junya Hoshi**

Tokyo Metropolitan Research Institute for Environmental Protection 1-7-5, Sinsuna, Koto-ku, Tokyo 136-0075, Japan; okada-m@tokyokankyo.jp (M.O.); funakubo-c@tokyokankyo.jp (C.F.); hoshi-j@tokyokankyo.jp (J.H.)

*** Correspondence: hata-h@tokyokankyo.jp

**Abstract:** High concentrations of tropospheric ozone remain a concern, and strategies to reduce the precursors of ozone, volatile organic compounds (VOCs) and nitrogen oxides, have been established in many countries. In this study, chassis dynamometer experiments were conducted for 25 late model gasoline passenger vehicles in the Japanese market to evaluate VOC emission trends. Tailpipe emissions were collected and analyzed using gas chromatography mass spectrometer and flame ionization detector, and liquid chromatography–mass spectrometry (LC-MS). Results showed that tailpipe VOC emissions increased linearly with vehicle mileage due to deterioration of the three-way catalysis converter used to purify the toxic components in vehicle emissions. Distance normalized total VOC emissions showed that port injection mini-sized vehicles were effective in decreasing tailpipe VOC emissions because of their low vehicle weight. The VOC composition of tailpipe emissions was dependent on the fuel type (regular or premium gasoline). VOC emissions from hybrid vehicles were similar to those of other vehicles because cooling of the three-way catalysis converter during battery operations sometimes tended to reduce catalyst effectiveness during engine operations. However, it can also be assumed that each manufacturer is aware of this phenomenon and is taking action. Further monitoring of hybrid vehicles is warranted to ensure that these vehicles remain an effective means of mitigating air pollution.

**Keywords:** chassis dynamometer; gasoline passenger vehicle; VOCs; three-way catalysis converter; Japan

---

## 1. Introduction

Levels of tropospheric ozone are a concern for many countries, and many nations have established regulations focused on ozone reduction [1–3]. For example, the United States and European Union have set maximum allowable ozone concentrations of 70 and 60 ppb (eight hourly average), respectively [4,5], while the Japanese government established a stricter standard (60 ppb hourly average) [6]. There are 1463 monitoring stations in Japan; however, ozone concentrations did not meet the environmental standard in 99.9% of them during FY2017 (fiscal year starting 1 April 2017) and a strategy to reduce ozone concentration has been demanded. Tropospheric ozone is generated by a photochemical reaction between nitrogen oxide ($NO_x$) and volatile organic compounds (VOCs) [7]; therefore, ozone abatement should focus on decreasing the levels of those compounds.

Previous studies have suggested that the atmospheric conditions driving tropospheric ozone generation can be either VOC or $NO_x$ sensitive [8–10]. Under a VOC sensitive regime, the amount of ozone generation is proportional to VOC emissions in the local region and inversely proportional to $NO_x$ emissions. VOC sensitive regimes mainly correspond with metropolitan and urban areas. Conversely, $NO_x$ sensitive regimes correspond with suburban and rural areas and ozone generation is proportional to local $NO_x$ emissions. While there are many $NO_x$ and VOCs emission sources, passenger

vehicles are thought to be the major contributor. Previous studies have assumed that passenger vehicles dominated in Japanese metropolitan and urban areas (VOC sensitive regime) due to their high populations; hence, they concluded that reducing VOC emissions from vehicles would be an effective means of controlling tropospheric ozone in metropolitan and urban areas [11]. Apart from their impact on tropospheric ozone, VOCs are also secondary organic aerosol (SOA) precursors [10,12] and are as harmful as tropospheric ozone for humans and other animals. There are two VOC emission mechanisms from vehicles; engine exhaust (tailpipe) and evaporative emissions. Tailpipe emissions are exhaust gases from engine combustion while the vehicle is running. Many reduction technologies for tailpipe emissions have been introduced. For example, three-way catalysts are used in gasoline cars [13] and selective catalytic reduction (SCR) or $NO_x$ storage reduction catalysts are used in diesel cars [14]. These technologies have successfully reduced $NO_x$, VOCs, and CO from tailpipe emissions; however, emission detoxification can be further improved. Evaporative emissions occur during long-term parking or refueling of gasoline vehicles [15,16] due to the high volatility of gasoline fuel. In Japan, more than 90% of passenger vehicles are gasoline, so it is important to address evaporation-related VOC emissions. To prevent gasoline evaporation, charcoal canisters are attached to vehicles to adsorb evaporative emissions during long-term parking. To reduce refueling emissions, on-board refueling emission traps, referred to as onboard refueling vapor recovery [17], are required on each gasoline vehicle in the United States. Stage 2 systems, which are evaporation prevention devices used in gasoline dispensing facilities, have also been introduced in Europe and some East Asian countries, but not in Japan [18].

Previous studies have described the detailed behavior and analytical methods of evaporative emissions from both long-term parking and refueling processes [19–21]; however, few studies have analyzed tailpipe VOC emissions from late model passenger vehicles in the Japanese market. To address this, we conducted chassis dynamometer measurements for late model gasoline passenger vehicles (mini-sized gasoline vehicles, standard-sized gasoline vehicles, and hybrid gasoline vehicles) in the Japanese market to understand VOC emission trends for each type of vehicle.

The purpose of this study is twofold. Firstly, we aim to share scientific discoveries and Japanese passenger vehicle VOC emission data with other countries to compare the effectiveness of VOC mitigation strategies worldwide. Secondly, information on late model Japanese market gasoline passenger vehicles will be used to create a new emission inventory, which can be used for chemical transport modeling in Japan. All of the VOC composition data are available as Supplementary Material.

## 2. Methods

### 2.1. Chassis Dynamometer Experiments for Late Model Gasoline Passenger Vehicles

In this study, gasoline passenger vehicles in the Japanese market were classified into five categories: port injection mini-sized vehicle (PI-m), port injection standard-sized vehicle (PI), direct injection vehicle (DI), direct injection premium gasoline vehicle (DI-p), and hybrid vehicle (HV). PI-m comprise small sized, minimum material vehicles with a displacement less than 0.66 L; a specific category in the Japanese market. PI-m, PI, DI, and HV are the main vehicle types in Japan, accounting for approximately 90% of all passenger vehicles. There are limited numbers of the sportier-type DI-p in the Japanese market. The gasoline used in the vehicle in Japanese market are classified into two types: regular and premium. Regular gasoline is fuel with octane numbers of at least 89 and premium gasoline is fuel with octane numbers of at least 96 [22].

Tailpipe exhaust emissions from each vehicle were measured using a chassis dynamometer (MEIDACS-DY6000, Meidensha Corporation). Experiments were conducted for 25 vehicles and information on these vehicles is shown in Table 1. Exhaust gas was diluted with dried air in the atmosphere and trapped using a sampling bag. Then, the concentrations of carbon monoxide (CO), carbon dioxide ($CO_2$), and total hydrocarbon (THC) were measured by an Automotive Emissions Analyzer (MEXA7400D, HORIBA); CO and $CO_2$ were measured using non-dispersive infrared

spectroscopy and THC was measured using a flame ionization detector (FID). Fuel consumption of each vehicle was calculated using the carbon balance method based on the concentrations of CO, $CO_2$, and THC. Nitrogen oxides for NO and $NO_x$ were measured using the chemiluminescence method and $N_2O$ was measured using non-dispersive infrared spectroscopy; however, these compounds are beyond the scope of this study. Exhaust gas was sampled to conduct a composition analysis of non-oxidized and oxidized VOCs, and details are described in the following subsection. The driving mode used in this study was JC08 [23], which was an approved test mode in Japan until 2018. The JC08 driving cycle was based on typical Japanese driving patterns on public roads and express highways; therefore, exhaust emissions measured in JC08 were assumed to be representative of average emissions from light-duty vehicles in Japan. The time profiles of the test mode are described in Figure A1 in Appendix A. All tests were conducted at a temperature of 25 ± 5 °C and 30%–75% humidity. Experiments were conducted from May 2015 to December 2018.

**Table 1.** Details of the vehicles tested in this study (R-gasoline: Regular gasoline, P-gasoline: Premium gasoline). The displacement, inertial weight, and mileage before the test exhibit the range of the tested vehicles.

| Vehicle Type | PI-m | PI | DI | DI-p | HV |
|---|---|---|---|---|---|
| Number of tested vehicles | 4 | 7 | 4 | 4 | 6 |
| Fuel type | R-gasoline | R-gasoline | R-gasoline | P-gasoline | R-gasoline |
| Displacement (L) | 0.658–0.659 | 1.242–2.359 | 1.496–1.997 | 1.490–1.997 | 1.496–2.493 |
| Inertial weight (kg) | 910–1020 | 1020–1930 | 1250–1700 | 1130–1590 | 1250–2150 |
| Mileage before test (km) | 15131–74204 | 1363–49810 | 358–16067 | 8053–38593 | 1586–53814 |

### 2.2. Composition Analysis of Non-Methane Hydrocarbons (Non-Methane VOCs)

The exhaust gas captured in the sampling bag was trapped using a high-vacuum 1 L stainless canister (Silonite 1 L MiniCan, Entech Instruments Inc. New York, United States) and diluted to 100 kPa using high purity $N_2$ gas. The composition of 55 non-oxidized VOCs was analyzed and determined by gas chromatograph mass spectrometer and flame ionization detector (GC-MS/FID). VOC measurements from May 2015 to March 2017 were conducted using GC-MS/FID from Agilent Technologies Inc. (GC: 7890 GC system; mass spectrometer: 5975C inert MSD California, United States), and measurements from April 2017 to December 2018 were conducted using a GC-MS/FID from Shimadzu Corporation (GCMS-QP2020 Kyoto, Japan).

The exhaust gases from the sampling line for each vehicle were collected using a DNPH solid-phase extraction cartridge (GL InertSep mini AERO DNPH, GL Science Co. Tokyo, Japan) with a portable gas sampling pump (GSP-2LFT, GASTEC Corporation Kanagawa, Japan) for solid-phase extraction. The composition of 14 aldehydes and ketones was measured using liquid chromatograph mass spectrometer (LC-MS, G6120B Quadrupole LC/MS, Agilent Technologies Inc. California, United States).

The total VOC amount for each vehicle was defined as the sum of the VOCs analyzed in this study. The detailed experimental setup of the GC-MS/FID and LC-MS are shown in Tables A1–A3 in Appendix A.

### 3. Results and Discussion

#### 3.1. Trends of Tailpipe VOCs Emissions and Ozone Formation Potential for Hot- and Cold-Starts

Figure 1 shows total amounts of VOC emissions from the 25 vehicles analyzed in this study following a hot- and cold-start. VOC emissions from PI-m vehicles appear higher as compared to other gasoline vehicles following both hot- and cold-starts. We attribute this to deterioration of the three-way catalyst (TWC) attached to the vehicles due to high mileage, and further details are provided in Section 3.2. The ratio of aromatic compounds in DI-p is higher than other gasoline vehicles. Premium gasoline is high octane fuel, meaning that the gasoline includes high levels of aromatic compounds compared to regular gasoline. The high ratio of aromatic compounds in DI-p emissions stems from

the incomplete combustion of the aromatics found in premium gasoline. Cold-start VOC emissions are approximately 30 times higher than hot-start emissions; meaning that a 30 km drive following a hot-start produces the same volume of emissions as a 1 km drive following a cold-start. On average, Japanese drivers travel approximately 30 km in each day [24], and for this reason, cold-start emissions are more important when managing VOC emissions from passenger vehicles than hot-start emissions. VOC composition data measured in this study are available in the Supplementary Material.

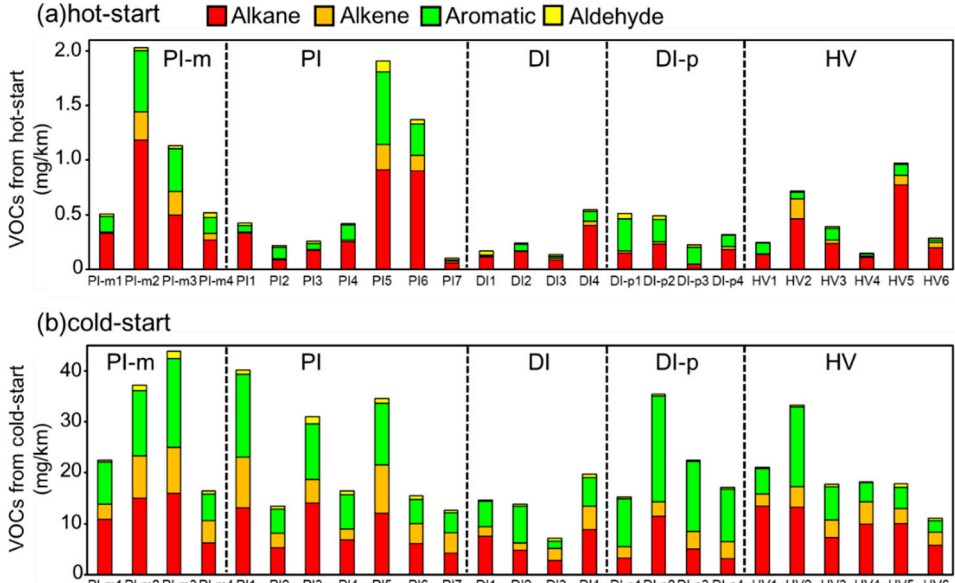

**Figure 1.** Total volatile organic compound (VOC) emissions for the 25 vehicles measured in this study following a (**a**) hot-start and (**b**) cold-start. (PI-m: mini-sized port injection, PI: standard-sized port injection, DI: direct injection, DI-p: direct injection premium gasoline, and HV: hybrid vehicles).

Figure 2 shows the ozone formation potential (OFP) of 25 vehicles evaluated according to the VOC emissions results in Figure 1. The equation is as follows:

$$\text{OFP}^j = \sum_i \text{MIR}_i \times \left[ \text{VOC}_i^j \right], \tag{1}$$

where MIR, *i*, and *j* are maximum incremental reactivity [25], VOC component *i*, and VOC type (alkane, alkene, aromatic, aldehyde), respectively. The OFP values are, on average, four to five times higher than total amount of VOCs; however, the trend for each vehicle between OFP and total amount of VOCs is almost similar. In general, alkenes, aromatics, and aldehydes have high MIR values, so vehicles that include a high amount of these compounds—such as PI-m2, PI-m3, PI5, and almost all cold-start emissions—contribute strongly to ozone formation in the atmosphere. As mentioned in the Introduction, there are two mechanisms for VOC emissions from gasoline cars: tailpipe exhaust and evaporative emissions. According to previous studies [19,21], the latter include a high ratio of alkanes. However, this study showed that tailpipe exhaust emissions include mainly high MIR compounds like alkenes and aromatics; thus, the management of tailpipe VOC emissions are still an important factor to address in terms of the formation of tropospheric ozone.

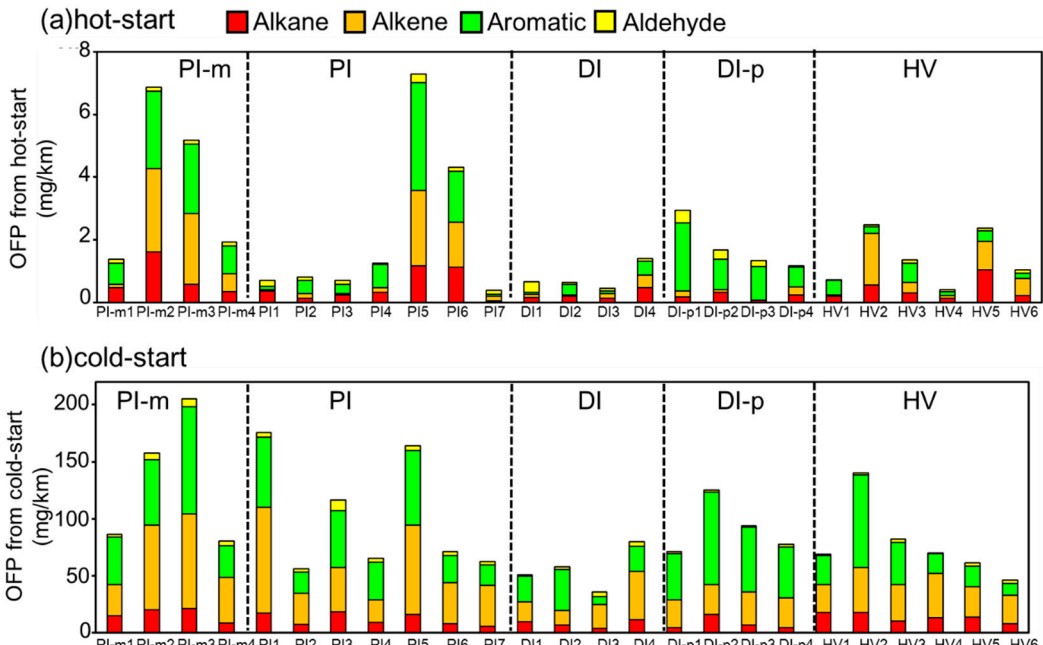

**Figure 2.** Ozone formation potential (OFP) for the 25 vehicles measured in this study following a (**a**) hot-start and (**b**) cold-start. (PI-m: mini-sized port injection, PI: standard-sized port injection, DI: direct injection, DI-p: direct injection premium gasoline, and HV: hybrid vehicles).

### 3.2. Increased VOCs from Tailpipe Emissions Caused by the Deterioration of the Three-Way Catalyst

According to Matsunaga et al. [26], a TWC attached to a gasoline vehicle will deteriorate with vehicle usage due to sintering or poisoning of the rare metal used in the catalyst. In other words, VOCs from tailpipe emissions can be expected to increase as the mileage of the vehicle increases. To confirm the effects of TWC deterioration on VOC emissions, two statistical approaches were applied to the total VOC emissions. First, the total VOC amount from vehicle $i$, $(A_i)$ was normalized by the weight of inertia ($w_i$, kg) to determine the weight-normalized total VOC amount $AN_i$ (mg/(km ton)) as follows:

$$AN_i = \frac{A_i}{w_i}. \tag{2}$$

This normalization was conducted because VOC and other tailpipe emissions are presumed to depend on vehicle weight, so this factor should be eliminated to fully understand the effects of TWC deterioration. The relationship between mileage and normalized total VOC emissions for each vehicle is shown in Figure 3. Total VOCs emissions increase as a linear function of the total mileage for each vehicle, and we attribute this to the deterioration of the TWC. The relationship between weight normalized VOC ($AN_i$) and mileage $d$ (km) for hot- and cold-starts (Figure 3) is calculated using linear regression as follows:

$$AN_i^{\text{hot}} = 1.9 \times 10^{-5}d + 0.04, \tag{3}$$

$$AN_i^{\text{cold}} = 4.2 \times 10^{-4}d + 5.3. \tag{4}$$

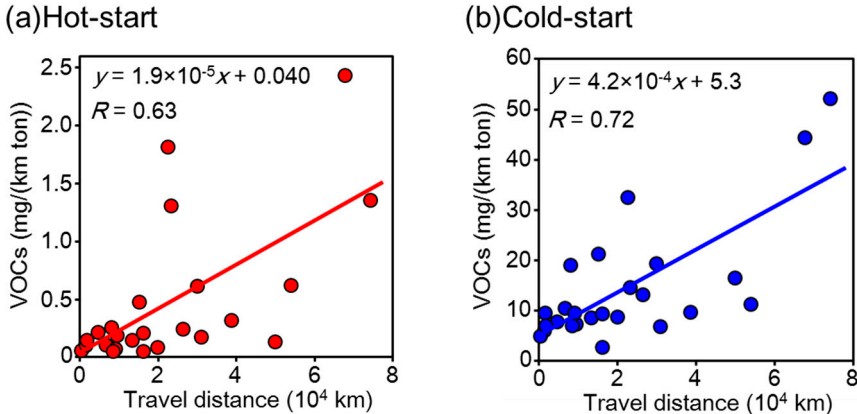

**Figure 3.** Relationship between mileage and total VOC emissions from the 25 vehicles observed in this study following a (**a**) hot-start and (**b**) cold-start.

Plot error can be attributed to factors including engine system properties and fuel consumption characteristics. Notwithstanding, results clearly show that vehicle VOC emissions are predominantly influenced by vehicle mileage and in this discussion, the VOC emissions calculated from Equation (2) are considered to be a function of mileage. In other words, if the mileage of each vehicle in the study was the same, then VOC emissions $A_i$ (mg/km) are assumed to be a function of vehicle weight, $w_i$ (ton), only. This suggests that when managing VOC emission trends, vehicle mileage should be an important consideration.

Moulijn et al. [27] suggested that TWC deterioration depended on the length of time that pollutants flowed into the catalyst according to:

$$\frac{dR}{dt} = k_d a^m, \tag{5}$$

where $R$ is the deterioration amount, $a$ is the catalyst activity, $m$ is the dimensionless parameter, and $k_d$ is the deterioration rate constant. This suggests that the rate of deterioration decreases with increased TWC activity, and that deterioration plateaus following long-term exposure to exhaust gas. Figure 3 shows the linear relationship between total VOC emissions and mileage, where mileage corresponds to time in Equations (3) and (4). However, in high mileage vehicles (>80,000 km), the slope of total VOC emissions as a function of mileage is expected to decrease. Despite this fact, Figure 3 suggests that VOCs emissions will increase almost linearly with mileage during normal vehicle use because the average mileage within the lifespan of the vehicle is approximately 100,000 km [24].

### 3.3. VOCs Emissions from Passenger Vehicles Normalized for Mileage

To compare the emission trends of each vehicle without the impact of TWC deterioration, VOC emissions from each vehicle, normalized for mileage, are calculated using Equations (6) and (7) for hot- and cold-starts, respectively:

$$A_i^{\text{hot}} = \left(1.9 \times 10^{-5}d + 0.04\right)w_i, \tag{6}$$

$$A_i^{\text{cold}} = \left(4.2 \times 10^{-4}d + 5.3\right)w_i. \tag{7}$$

The inertial weights ($w_i$) for each vehicle are listed in Table S1 in the Supplementary Material. The average mileage of passenger vehicles in Japan is approximately 60,000 km [24]. Using this information, and assuming that the VOC composition ratio does not change with mileage, the normalized average VOC emissions for each vehicle type are shown in Figure 4. It is apparent that PI-m emit less VOC compared to other vehicle types because of their light weight. The overall emission amounts are approximately equal for the other vehicle types. DI-p vehicle emissions include a high ratio of aromatic compounds compared to other vehicles. Aromatic compounds usually have a high maximum

incremental reactivity for ozone [25], so these vehicles are thought to be high contributors to ozone formation. It is notable that VOC emissions from HVs are not lower than other vehicle types and this is discussed further in the following section. The results analyzed in Figure 4 include the assumption that the increase in VOCs depends only on mileage; other factors are neglected. The exhaust emitted from vehicles VOCs that are expected to be proportional to variables such as the structure and material of the TWC, the position where it is attached, and the record of how vehicles were used (e.g., in urban or rural areas). The analysis in Figure 4 should be made further sophisticated, in terms of the uncertain factors mentioned above, in future studies.

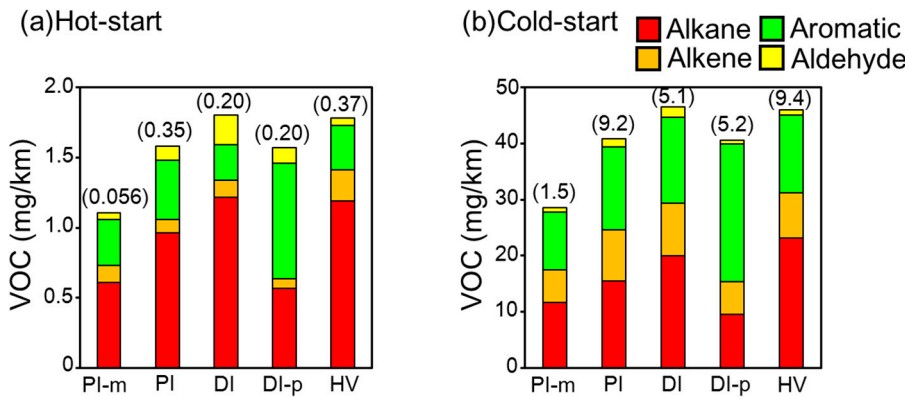

**Figure 4.** Normalized average VOC emissions by each vehicle type for a hot-start (**a**) and cold-start (**b**). Values in parentheses represent the standard deviation for each vehicle type.

### 3.4. High VOC Emissions from a Specific Hybrid Vehicle

While HV are expected to be low emission vehicles because of their engine battery combined driving process, results of this study indicate that HV emissions are approximately equal to other vehicle types. This anomaly is the result of a single vehicle (HV2) with very high VOCs emissions. Figure 5 shows time profiles of cold-start non-methane hydrocarbons (NMHC; equal to total VOC), engine rotation, and exhaust gas temperature for HV2 and a reference vehicle, PI4, which represents the generally observed distribution of VOC emissions. The NMHC emission-time profile of PI4 shows that high amounts of NMHC are emitted in the first few minutes of driving and then emissions reduce markedly over time. This pattern can be explained by the time-exhaust gas temperature profile in Figure 5a; temperatures in the first few minutes are low due to less reaction enthalpy from the chemical reaction in the TWC. After a few minutes of driving, the catalyst was warmed by engine exhaust gases and the detoxifying performance of the TWC was improved. Continuous NMHC emissions from HV2 following a cold-start (Figure 5b) show smaller initial peak emissions and emission spikes over time. This is caused by the vehicle's dual driving processes; intermittent engine use enables the temperature of the TWC to reduce, impacting its effectiveness. However, this pattern was not observed in other HV. Hybrid car manufacturers have adopted countermeasures to limit continuous pollutant emissions during engine operations by placing the TWC close to the engine to reduce the catalyst warm up time. Notwithstanding, our results indicate that high NMHC emissions can sometimes occur from hybrid vehicles due to TWC cooling during battery operations. However, it can also be assumed that each car manufacturer has already noticed this issue and may be taking steps regarding this phenomenon. Hence, it is important to monitor NMHC emissions over time for the various hybrid vehicles on the market.

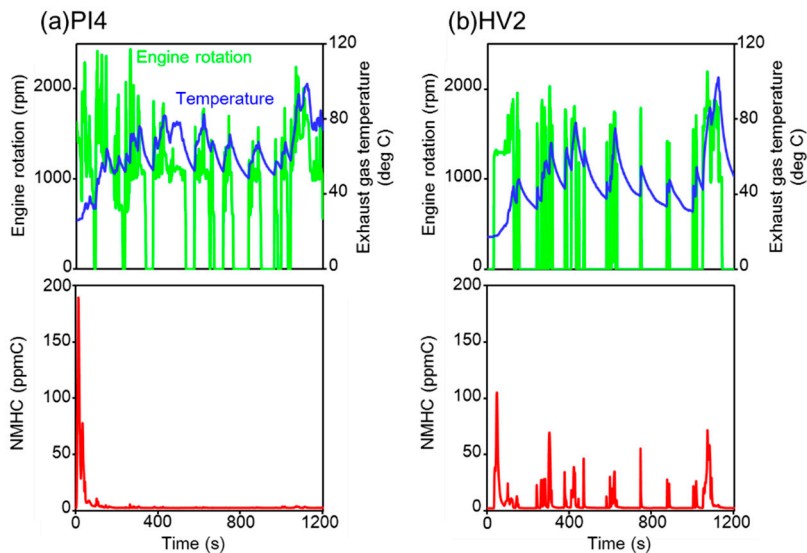

**Figure 5.** Relationship between fuel consumption and VOC emissions from vehicles PI-4 (**a**) and HV2 (**b**) following a cold start.

## 4. Conclusions

VOC emissions from five types of gasoline vehicles (PI-m, PI, DI, DI-p, and HV) were measured using chassis dynamometers, and detailed chemical components were determined using GC-MS/FID and LC-MS. Results showed that PI-m vehicles emitted high total VOCs following both hot- and cold-starts due to the high mileage of the vehicles tested in this group. Emissions from DI-p included a high ratio of aromatics because the premium gasoline used in these vehicles includes a high ratio of aromatic compounds including benzene and toluene. The trend in the ozone formation potential calculated for the measured vehicles was almost the same as the trend in VOCs emissions for each vehicle. Total VOC emissions were approximately proportional to mileage due to the deterioration of the TWC over time. Distance normalized total VOC emissions showed that mini-sized PI-m vehicles were effective in decreasing tailpipe VOC emissions because of their low vehicle weight. Hybrid vehicles were found to be less effective in reducing tailpipe VOC emissions because TWC tended to cool during battery operations, decreasing the effectiveness of the catalyst during engine operations. Gasoline vehicles remain the most common passenger vehicle type worldwide, and the results of this study will help to guide new regulations to facilitate the successful reduction of tropospheric ozone from these vehicles.

**Supplementary Materials:** VOC composition data for the 25 vehicles used in this study are available online at http://www.mdpi.com/2073-4433/10/10/621/s1.

**Author Contributions:** H.H. wrote this paper and engaged in chassis dynamometer experiments and analysis of the experimental results. M.O. and C.F. conducted the VOC composition analysis using GC-MS/FID and LC-MS. J.H. was the leader of chassis dynamometer experiments.

**Acknowledgments:** This research was financially supported by Bureau of Environment, Tokyo Metropolitan Government.

**Conflicts of Interest:** The authors declare no conflicts of interest.

## Appendix A

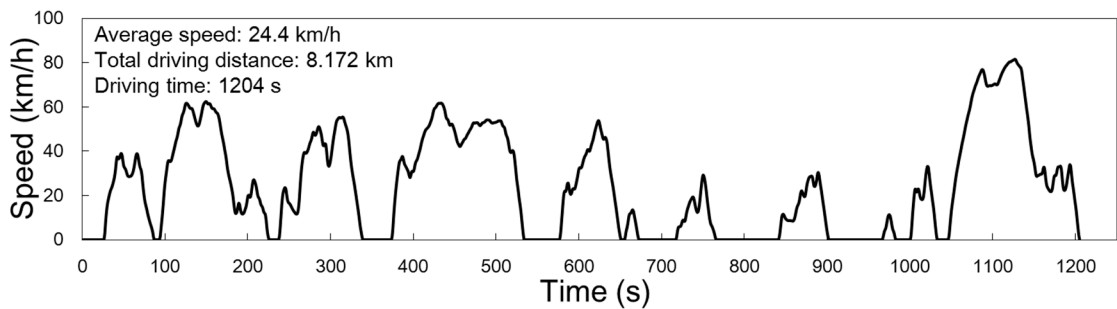

**Figure A1.** Summary of JC08 driving test mode, the approved vehicle test mode used in Japan until 2018.

**Table A1.** Experimental setup for gas chromatography–flame ionization detection (GC-FID) used in this study.

| Column Temperature rising Program | | |
|---|---|---|
| | Type | TC-BOND Alumina/KCl (50 m, 0.53 mmID, 10 um) |
| | Initial temp. (°C) | 40 |
| | Initial temp. retention time (min.) | 5 |
| | First temp. rate of increase (°C/min.) | 2.5 |
| | First retention temp. (°C) | 130 |
| | First retention time (min.) | 0 |
| | Second temp. rate of increase (°C/min.) | 20 |
| | Final retention temp. (°C) | 200 |
| | Final retention time (min.) | 15.5 |
| Carrier gas | Type | He |
| | Pressure (kPa) | 400 |
| Detector | Type | FID |
| | Temp. (°C) | 250 |
| | Fuel gas (mL/min.) | H2 40 |
| | Combustion gas (mL/min.) | Air 400 |
| | Additional gas (mL/min.) | N2 30 |

**Table A2.** Experimental setup of gas chromatography–mass spectrometry (GC-MS).

| Column Temperature riding Program | | |
|---|---|---|
| | Type | DB-1 (60 m, 0.25 mmID, 1 um) |
| | Initial temp. (°C) | 40 |
| | Initial temp. rate of increase (min.) | 5 |
| | First temp. rising velocity (°C/min.) | 3 |
| | First retention temp. (°C) | 180 |
| | First retention time (min.) | 0 |
| | Second temp. rate of increase (°C/min.) | 30 |
| | Final retention temp. (°C) | 250 |
| | Final retention time (min.) | 6 |
| Carrier gas | Type | He |
| | Pressure (kPa) | 400 |
| Detector | Type | Shimadzu GCMS-QP2020 |

**Table A3.** Experimental setup of liquid chromatography–mass spectrometry (LC-MS).

| Column Temperature riding Program | Type | Poroshell 120 EC-C18 (2.1×100 mm, 2.7 um) |
|---|---|---|
| | Mobile Phase A<br>Mobile Phase B | 95:5 (v/v) water / acetonitrile<br>acetonitrile |
| | Gradient | Time (min)　　　　B (%)<br>0　　　　　　　　　20<br>2　　　　　　　　　20<br>8　　　　　　　　　40<br>12　　　　　　　　50<br>22　　　　　　　　60<br>23　　　　　　　　100 |
| | Flow rate<br>Column temp. (ºC) | 0.4 mL/min<br>40 |
| **Injection part** | Volume (uL)<br>wash solvent<br>Injection amount (uL) | 100<br>acetonitrile<br>5 |
| **Detector** | Type | Agilent 6120API-ES, Negative, SIM |

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
