# Peer review of "Tailpipe VOC Emissions from Late Model Gasoline Passenger Vehicles in the Japanese Market"

_atmosphere, doi:10.3390/atmos10100621_

Round 1

Reviewer 1 Report

This paper presented some interesting findings regarding the tailpipe VOC emissions from gasoline passenger vehicles in Japan. The authors conducted a comprehensive investigation of 5 major passenger vehicle types and found that port injection mini-size vehicle was most effective in decreasing tailpipe VOC. The authors also found that hybrid vehicles are not necessarily to be more efficient in reducing VOC emissions, which I think would be an important spotlight for this field. The paper is well written and provided solid methodology and analyses. I think what is found in this paper would be very interesting to the journal readers. I would recommend it for publication after minor revision.

Minor comments:

In the introduction, the author only emphasized the impact of VOC on ozone. However, VOC also plays an important role in secondary aerosol formation. I would recommend the authors to add some related citations so that this study could have a border impact. such as: https://www.sciencedirect.com/science/article/pii/S1352231018301158 and https://www.sciencedirect.com/science/article/pii/S1352231019303206 Page 2 Line 70: the author state that the purpose of this study is to compare the effectiveness of VOC mitigation strategies worldwide. However, I could not find related analyses in the following manuscript. Page 2 Line 95, it would be better to add a citation for the JC08 driving mode. Table 1. The author should provide a citation or detailed definition for "Regular Gasoline" and "Premium Gasoline" Page 6 Line 191, The author should provide citation for the claim: "aromatic compounds usually have a high maximum incremental reactivity for ozone".

Author Response

Comments and Suggestions for Authors
This paper presented some interesting findings regarding the tailpipe VOC emissions from gasoline passenger vehicles in Japan. The authors conducted a comprehensive investigation of 5 major passenger vehicle types and found that port injection mini-size vehicle was most effective in decreasing tailpipe VOC. The authors also found that hybrid vehicles are not necessarily to be more efficient in reducing VOC emissions, which I think would be an important spotlight for this field. The paper is well written and provided solid methodology and analyses. I think what is found in this paper would be very interesting to the journal readers. I would recommend it for publication after minor revision.
(Answer)
The authors feel gratitude for the positive opinion and important suggestions from reviewer1. We revised the manuscript carefully according to your suggestions. All the improved sentences are drawn with yellow.
Minor comments:
In the introduction, the author only emphasized the impact of VOC on ozone. However, VOC also plays an important role in secondary aerosol formation. I would recommend the authors to add some related citations so that this study could have a border impact. such as: https://www.sciencedirect.com/science/article/pii/S1352231018301158 and https://www.sciencedirect.com/science/article/pii/S1352231019303206
(Answer1)
Thank you for the critical suggestion to the introduction and recommendation of the references. We agree with the opinion about the role of VOCs to SOA formation, and added as following sentence in line 49-51based on the suggested articles.
Apart from their impact on tropospheric ozone, VOCs are also secondary organic aerosol (SOA) precursors [10,12] and are as harmful as tropospheric ozone for humans and other animals.
Page 2 Line 70: the author state that the purpose of this study is to compare the effectiveness of VOC mitigation strategies worldwide. However, I could not find related analyses in the following manuscript.
(Answer2)
This sentence means that the VOC emission data in supplementary material is useful to compare VOC emission trend in the different countries.
Page 2 Line 95, it would be better to add a citation for the JC08 driving mode.
(Answer3)
We agreed with your opinion and added the citation of JC08 (Reference 22).
 Table 1. The author should provide a citation or detailed definition for "Regular Gasoline" and "Premium Gasoline" Page 6 Line 191, The author should provide citation for the claim: "aromatic compounds usually have a high maximum incremental reactivity for ozone".
(Answer4)
According to your suggestion, the definition of regular and premium gasoline was added as following sentence in line 87-88. Also the citation of MIR was added in the corresponding line.
Regular gasoline is fuel with octane numbers of at least 89 and premium gasoline is fuel with octane numbers of at least 96 [22].

Reviewer 2 Report

Review of the manuscript « Tailpipe VOC emissions from late model gasoline passenger vehicles in the Japanese market » by Hata et al.. The manuscript describes chassis dynamometer experiments with 25 gasoline powered vehicles representative of the Japanese market, namely port injector engine of mini and regular sizes, direct injection with regular and premium gasoline and hybrid vehicles. Results show strong heterogeneity and certain trends, as expected. Overall I find the experiments robust and most of analysis are convincing. There are certain aspects of the analysis that I do not agree, and certain points that could be pushed just a little bit further. The manuscript should be accepted after the following points are addressed:

General comments:

The analysis of the impact of deterioration of TWC with mileage is too simplistic. In the very least, it is lacking the uncertainty of the fit parameters (which I think are very large), and their propagation into the other estimates (which will be substantial). If the authors wish to keep this analysis, uncertainties need to be accounted. However, my feeling is that the statistics are not large enough for this type of assessment to be done (the same model had to be followed at different mileages, and individual VOC species or groups be followed). Therefore, I do not think authors should not use this to base their conclusions. Instead, this subsection could be replaced by the following point. Given the motivation being O3 formation, which has been well put in the introduction, I’d suggest the authors to replace the TWC deterioration with an analysis of ozone forming potential by vehicular category, without normalization whatsoever. This is another way of looking at the speciated VOC emission with direct impact on air quality. This could be separated by weight categories, or by engine type, and split between hot and cold start in the form of pie charts. I find interesting the discussion with hybrid vehicles, however this would need to have an improved contextualization. Those vehicles are typically heavier, and their VOC emission is not so significantly smaller than a more compact vehicle (the same as CO2, actually, for those that are not plug-in). However, it seems that one of the models does not control temperature correctly (HV-2) and thus has higher emission during the testing cycle. This temperature issue seems to be something that constructors are aware of, and seems to me at least less important than the abstract led to believe. This needs to be more accurately written, particularly in the abstract, I believe.

Specific comments:

23 “warranted to ensure” 33-L.34 Re-word somewhat this sentence (perhaps use 99.9% instead of 0.1% makes it clearer) 34 Also “FY2017” is not clear for the reader 36-38 “ (NO x ) and volatile organic compounds (VOCs) [7], so its ozone abatement should focus on decreasing those compounds”. 69: There are other venues of data sharing, the study has to focus on scientific discoveries. Please re-word accordingly. 95 Can please the authors describe a bit more what the follow up of JC08 can impact on the newer studies? How will they compare with the results shown here? 132 “in this study is available”. 3: Main text says it is normalized, but not on the caption.

Author Response

(Answer for reviewer 2)
Comments and Suggestions for Authors
Review of the manuscript « Tailpipe VOC emissions from late model gasoline passenger vehicles in the Japanese market » by Hata et al.. The manuscript describes chassis dynamometer experiments with 25 gasoline powered vehicles representative of the Japanese market, namely port injector engine of mini and regular sizes, direct injection with regular and premium gasoline and hybrid vehicles. Results show strong heterogeneity and certain trends, as expected. Overall I find the experiments robust and most of analysis are convincing. There are certain aspects of the analysis that I do not agree, and certain points that could be pushed just a little bit further. The manuscript should be accepted after the following points are addressed:
(Answer)
The authors feel gratitude for the positive opinion to our article. According to your suggestion, the manuscript was carefully modified. All the improved sentences are colored with yellow.
General comments:
The analysis of the impact of deterioration of TWC with mileage is too simplistic. In the very least, it is lacking the uncertainty of the fit parameters (which I think are very large), and their propagation into the other estimates (which will be substantial). If the authors wish to keep this analysis, uncertainties need to be accounted. However, my feeling is that the statistics are not large enough for this type of assessment to be done (the same model had to be followed at different mileages, and individual VOC species or groups be followed). Therefore, I do not think authors should not use this to base their conclusions.
(Answer1)
Thank you for pointing out the luck of information about our discussion. We are also understanding that there are the uncertainty for the TWC deterioration discussion which was parameterized only by the mileage. In this study, we could not proceed further analysis because of the luck of the information about critical parameters related to VOC emission (e.g. previous driving pattern of each vehicle, TWC structure etc.). This means that it is also difficult to numerically evaluate the uncertainty of the analyzed results. Despite this fact, it is clear that mileage is one of the most important factor to VOC emission from our experimental results. The uncertainty in our analysis could be included in the correlation factor in Figure 2. To our best knowledge, the previous researches related to chassis dynamometer conducted by a lot of researchers have not evaluated VOC emissions in terms of TWC deterioration and mileage. On the other hand, TWC deterioration researches were conducted by the specialists of catalysis. One of the purposes of our analysis is to collaborate these two research fields, and for this reason, we want to remain the discussion of VOC emission and TWC deterioration. Meanwhile, we strongly understand what the reviewer2 wanted to claim, the limitation of our analysis and the fact the further analysis should be conducted in the future are added in the article as below (lines 222-227). Thank you for your consideration.
The results analyzed in Figure 4 include the assumption that the increase in VOCs depends only on mileage; other factors are neglected. The exhaust emitted from vehicles VOCs that are expected to be proportional to variables such as the structure and material of the TWC, the position where it is attached, and the record of how vehicles were used (e.g., in urban or rural areas). The analysis in Figure 4 should be made further sophisticated, in terms of the uncertain factors mentioned above, in future studies.
Instead, this subsection could be replaced by the following point. Given the motivation being O3 formation, which has been well put in the introduction, I’d suggest the authors to replace the TWC deterioration with an analysis of ozone forming potential by vehicular category, without normalization whatsoever. This is another way of looking at the speciated VOC emission with direct impact on air quality. This could be separated by weight categories, or by engine type, and split between hot and cold start in the form of pie charts.
(Answer2)
Thank you for the suggestion above. As for the reason described in the former answer, the authors thought that TWC deterioration section should be remained. But also we agreed with the suggestion to include ozone formation potential (OFP) discussion in the manuscript. Therefore, this discussion was added in subsection 1 (lines 148-166 and Figure 2).
I find interesting the discussion with hybrid vehicles, however this would need to have an improved contextualization. Those vehicles are typically heavier, and their VOC emission is not so significantly smaller than a more compact vehicle (the same as CO2, actually, for those that are not plug-in). However, it seems that one of the models does not control temperature correctly (HV-2) and thus has higher emission during the testing cycle. This temperature issue seems to be something that constructors are aware of, and seems to me at least less important than the abstract led to believe. This needs to be more accurately written, particularly in the abstract, I believe.
(Answer3)
Thank you for the important suggestion of the abstract. The authors had also felt that the description of hybrid vehicles in abstract let the readers mislead sometimes, so the sentence was improved based on your insight (lines 23-24, 250-251).
(lines 23-24) However, it can also be assumed that each manufacturer is aware of this phenomenon and is taking action.
(lines 250-251) However, it can also be assumed that each car manufacturer has already noticed this issue and may be taking steps regarding this phenomenon.

Specific comments:
23 “warranted to ensure”
(Answer4)
Thank you to clarifying the mistake. The words were corrected.
33-L.34 Re-word somewhat this sentence (perhaps use 99.9% instead of 0.1% makes it clearer)
(Answer5)
According to the suggestion, the sentence was modified as below (line 34-35).
ozone concentrations did not meet the environmental standard in 99.9 % of them
34 Also “FY2017” is not clear for the reader
(Answer6)
Thank you to inform the unclearness of the word. FY means the fiscal year and the explanation was added in the sentence as below (line 35-36).
FY2017 (fiscal year starting April 1, 2017)
36-38 “ (NO x ) and volatile organic compounds (VOCs) [7], so its ozone abatement should focus on decreasing those compounds”.
(Answer7)
Thank you for the suggestion. The sentence was improved based on the suggestion (lines 38-39).
69: There are other venues of data sharing, the study has to focus on scientific discoveries. Please re-word accordingly.
(Answer8)
We agree with the suggestion and the sentence was modified based on your suggestion (line 72).
95 Can please the authors describe a bit more what the follow up of JC08 can impact on the newer studies? How will they compare with the results shown here?
(Answer9)
According to your suggestion, more detail about JC08 was added as the following sentence (lines 101-104).
JC08 driving cycle was based on typical Japanese driving patterns on public roads and express highways; therefore, exhaust emissions measured in JC08 were assumed to be representative of average emissions from light-duty vehicles in Japan.
132 “in this study is available”.
(Answer10)
Thank you for clarifying miss word. We modified it (line 142).
3: Main text says it is normalized, but not on the caption.
(Answer11)
Thank you for clarifying it. The term was added to the caption of Figure 4 (line 229).

Reviewer 3 Report

As a reviewer I have the following remarks

Table 1. Are these numbers (0.658-0.659) are ranges? Should be clarified. Figure 3. Is any reason to show (0.056) rather than (0.06). In general, the paper is well written and presented.

Thank you.

Author Response

(Answer for reviewer 3)
Comments and Suggestions for Authors
As a reviewer I have the following remarks
Table 1. Are these numbers (0.658-0.659) are ranges? Should be clarified. Figure 3. Is any reason to show (0.056) rather than (0.06). In general, the paper is well written and presented.
Thank you.
(Answer1)
The authors feel gratitude for the positive comments and important clarification for our article. As pointed out by reviewer3, the numbers such as (0.658-0.659) mean the range of the weight of vehicles, and according to your suggestion, the meaning of the numbers is described as following sentence in the caption in lines 109-110.
The displacement, inertial weight, and mileage before the test exhibit the range of the tested vehicles.
In this article, we treated the value from observed data based on the significant number "2" because of the accuracy of measurement system. Therefore, 0.056 is assigned in Figure 3.